# Chemical manipulation of an activation/inhibition switch in the nuclear receptor PXR

Efren Garcia-Maldonado[1,3], Andrew D. Huber [1,3] ✉, Sergio C. Chai [1,3], Stanley Nithianantham [1,3], Yongtao Li [1,3], Jing Wu[1], Shyaron Poudel[1], Darcie J. Miller[2], Jayaraman Seetharaman[2] & Taosheng Chen [1] ✉

Nuclear receptors are ligand-activated transcription factors that can often be useful drug targets. Unfortunately, ligand promiscuity leads to two-thirds of receptors remaining clinically untargeted. PXR is a nuclear receptor that can be activated by diverse compounds to elevate metabolism, negatively impacting drug efficacy and safety. This presents a barrier to drug development because compounds designed to target other proteins must avoid PXR activation while retaining potency for the desired target. This problem could be avoided by using PXR antagonists, but these compounds are rare, and their molecular mechanisms remain unknown. Here, we report structurally related PXR-selective agonists and antagonists and their corresponding co-crystal structures to describe mechanisms of antagonism and selectivity. Structural and computational approaches show that antagonists induce PXR conformational changes incompatible with transcriptional coactivator recruitment. These results guide the design of compounds with predictable agonist/antagonist activities and bolster efforts to generate antagonists to prevent PXR activation interfering with other drugs.

The human nuclear receptor (NR) superfamily contains 48 ligand-activated proteins that modulate diverse transcriptional targets and are involved in various physiological and pathological processes[1]. Due to their defined ligand binding domains (LBDs), NRs form a privileged family that is targeted by 16% of approved small molecule drugs[2]. This percentage is remarkable when the small number of NRs ($n = 48$) is compared to the overall size of the human proteome ($n = -20,000$)[3], and even more striking is that only approximately one-third of the NR family is targeted by approved drugs[4,5]. Untargeted NRs are implicated in a variety of diseases, but because of the lipophilic nature of NR ligands, there is considerable overlap of ligand-binding profiles, leading to undesirable off-target effects and hindering progress of additional NR-targeting drugs. Therefore, combinatorial approaches consisting of new chemical matter, biochemical and cellular evaluations, and structural studies are

required to successfully develop selective modulators with therapeutic value.

Of the clinically unutilized NRs, pregnane X receptor (PXR) is special in its broad potential applications that span all small molecule therapeutic categories. As a xenobiotic sensor, PXR is activated by structurally diverse compounds and plays a major role in drug metabolism and drug-drug interactions by regulating transcription of genes encoding drug metabolizing enzymes, drug conjugating enzymes, and drug transporters[6–9]. Perhaps the most well-known PXR target genes encode cytochrome P450 3 A (CYP3A) enzymes, which metabolize more than half of all clinical drugs[10], and modulation of CYP3A activity or expression levels are the dominant sources of drug-drug interactions[11]. PXR binding by various small molecules, including chemotherapies (e.g., paclitaxel)[12], antivirals (e.g., efavirenz)[13], antibiotics (e.g., rifampicin)[6–8], and environmental toxins[14,15], can compromise

[1]Department of Chemical Biology and Therapeutics, St. Jude Children's Research Hospital, 262 Danny Thomas Place, Memphis, TN 38105, USA. [2]Department of Structural Biology, St. Jude Children's Research Hospital, 262 Danny Thomas Place, Memphis, TN 38105, USA. [3]These authors contributed equally: Efren Garcia-Maldonado, Andrew D. Huber, Sergio C. Chai, Stanley Nithianantham, and Yongtao Li. ✉e-mail: andrew.huber@stjude.org; taosheng.chen@stjude.org

drug efficacy and/or safety. For example, PXR activators may reduce the effectiveness of co-administered oral contraceptives by enhancing their metabolism by PXR-regulated enzymes[16]. Analogously, inhibition of PXR can enhance the chemotherapeutic activity of paclitaxel by reducing paclitaxel metabolism[17]. In addition to impacting drug efficacy, PXR is involved in physiological responses such as hemorrhagic shock induced liver injury, and pharmacological PXR inhibition has proven beneficial in this context[18]. Thus, PXR antagonists as cotreatments have translational potential for diverse clinical indications. Importantly, drug cotreatments with metabolic modulators have highly successful precedents, such as the SARS-CoV-2 drug Paxlovid, which is a combination of a direct acting antiviral and the CYP3A inhibitor ritonavir[19].

Because of PXR's large, flexible ligand binding pocket that is evolutionarily tuned to activate the receptor[20,21], development of selective PXR antagonists has historically been deemed a considerable challenge[22]. However, using an unbiased high-throughput screening approach, we previously identified SPA70 as a potent and selective PXR antagonist with activity in vivo[17,23]. Surprisingly, subsequent structure-activity relationship studies revealed that subtle chemical changes convert SPA70 into an agonist[24], and further derivatization yielded potent compounds of both agonist and antagonist classes[25]. Because the previous work was performed through empirical medicinal chemistry efforts, and there are no crystal structures of antagonist-bound PXR, it is currently unknown how seemingly insignificant chemical modifications result in such drastic antagonist-to-agonist activity switches.

Here, we report structures of PXR LBD in complex with four antagonists and two structurally related agonists to describe a unified mechanism for agonism versus antagonism. We propose that our 1$H$−1,2,3-triazole-4-carboxamide scaffold is, in essence, a true antagonist scaffold that functions by reorienting residues of the PXR activation function-2 (AF-2) domain (most notably alpha helix 12, α12), thereby destabilizing the active α12 conformation. This destabilization results in a structural rearrangement that is incompatible with coactivator binding. Addition of a simple hydrophobic moiety (i.e., methoxy) at the α12-interacting region reintroduces stabilizing contacts between the ligand and α12, thereby resulting in a distinct mode of activation compared to other agonists. Furthermore, moieties directed at hydrophobic "hot spots" in other areas of the ligand binding pocket enhance binding affinity and cellular potency but do not impact the final biological outcome of agonist versus antagonist. These observations guided us to predict, design, and synthesize a series of chemical analogs with either agonistic or antagonistic profiles, all of which display high potency and PXR selectivity with minimal cytotoxicity. Our results pave the way for structure-guided generation of more effective PXR antagonists that may be used to prevent PXR-mediated metabolic liabilities.

## Results

### The ligand-α12 interface dictates PXR agonism and antagonism
Since the discovery of SPA70 and its analogs as PXR antagonists and agonists[17,23,24], we have described related compounds having an amide bond in lieu of the sulfonyl linkage found in SPA70, exemplified by SJPYT-310 (Fig. 1a) as a more potent PXR antagonist than SPA70[25]. Although there are >40 reported structures of PXR LBD bound to agonists[21], including the SPA70 analog SJB7 (Fig. 1a)[17], there is currently no structure of PXR with a bound antagonist, and it is unclear how binding of chemically similar compounds results in opposite transcriptional activities. To identify structural determinants of PXR antagonism, we obtained the crystal structure of SJPYT-310-bound PXR LBD, where the antagonist resides in the ligand binding pocket (Fig. 1b–d, Supplementary Fig. 1, Supplementary Table 1). The triazole ring of SJPYT-310 forms hydrogen bonds with Q285 (Supplementary Fig. 1a), the 2-methoxy-5-methylphenyl moiety (region A, Fig. 1a) faces

α12 (Fig. 1c), and the pentan-2-yloxy moiety from region B (Fig. 1a) is buried inside the hydrophobic "π-trap" consisting of F288, W299, and Y306 (Fig. 1d). On the opposite side of the region B phenyl ring, the *tert*-butyl moiety is fixed in a "leucine cage" composed of L206, L209, L239, and L240 (Fig. 1d).

A comparison among all available PXR LBD structures shows that SJPYT-310 binds in a scaffold-specific manner compared to all other ligands (Fig. 1e, Supplementary Fig. 1b), and this ligand position results in protein rearrangements in the AF-2 region not seen in previous structures. Specifically, region A of SJPYT-310 reorients F251, L428, and F429 to distinct positions (Fig. 1f). The F251 position is only shared by the related sulfonyl-based agonist SJB7 and is due to a clash between region A and the native F251 rotamer (Supplementary Fig. 1b, c). Importantly, the SJPYT-310-specific position of L428 loses SRC-1 interaction at L690, a key residue of the coactivator LXXLL NR-binding motif that is required for NR interaction (Fig. 1f)[26]. Thus, we have captured a PXR-antagonist-SRC-1 intermediate that may exist prior to antagonist-induced coactivator release. Alternatively, this may represent a means by which antagonists prime the AF-2 to prevent PXR interaction with coactivator.

To investigate differences between antagonist- and agonist-bound PXR LBD, we compared the structures of PXR LBD bound to SJPYT-310 or SJB7 (PDB ID 5XOR)[17] (Fig. 1g, Supplementary Fig. 1b–e). Although the compounds are chemically similar (Fig. 1a), there are differences in their binding modes (Supplementary Fig. 1b–e). Rather than triazole-Q285 hydrogen bonds, the sulfonyl group of SJB7 interacts with H407 through hydrogen bonding (Supplementary Fig. 1d). The *tert*-butyl moiety of SJPYT-310 (region B) is flipped 120˚ away from the π-trap and oriented toward the leucine cage, and the π-trap is instead occupied by the pentan-2-yloxy moiety, which is absent in SJB7 (Fig. 1g, Supplementary Fig. 1e). Binding of SJPYT-310 to both the π-trap and the leucine cage appears to stabilize alpha helix 2 (α2), which is disordered in PXR LBD structures with SJB7 and certain other agonists (Supplementary Fig. 1e)[17,21,27]. The increased region B interactions with both the π-trap and leucine cage likely account for the enhanced binding affinity of the amide series compared to the sulfonyl series[24,25] but may not be indicative of agonist/antagonist property. To specifically assess agonist/antagonist activities, we examined α12, which is a feature central to NR function[28,29]. Interestingly, although SJB7 induces F251, L428, and F429 reorientations, the 4-methoxy group of SJB7 region A likely promotes PXR activity by occupying a hydrophobic cleft formed by M425, L428, and F429 in α12 (Fig. 1g, sphere), stabilizing the helix conformation suitable for coactivator binding. This is evident in the proper positioning of L428 for SRC-1 contact (Fig. 1h, Supplementary Fig. 1c). SJPYT-310 lacks the 4-methoxy and cannot establish this interaction network (Fig. 1c), possibly resulting in an unfavorable surface for coactivator binding. To show that these results were not due to our protein construct or crystallographic conditions, we crystallized the apo form of PXR LBD, which is similar to previous apo structures[15,30–32] (Supplementary Fig. 2, Supplementary Tables 1 and 2). Taken together, we postulate that both the amide- and sulfonyl-linked 1$H$-1,2,3-triazole scaffolds are antagonistic by nature due to their distinctive binding positions and remodeling of the AF-2 region; however, agonistic behavior can be recovered by establishing specific ligand-α12 interactions through addition of a simple hydrophobic moiety (i.e., methoxy).

### Predictive design yields nontoxic PXR-selective modulators
Based on the structural analysis, we envisioned that analogs with a region A 4-methoxy group would behave as agonists, while those with a small 4-position group would prevent PXR's activation. We synthesized a set of SJPYT-310 analogs to test this prediction while varying additional moieties to assess effects on binding and activity (Fig. 2). Because of the presence of the SJPYT-310 *tert*-butyl moiety in the leucine cage, we designed SJPYT-331 with an ester group in place of

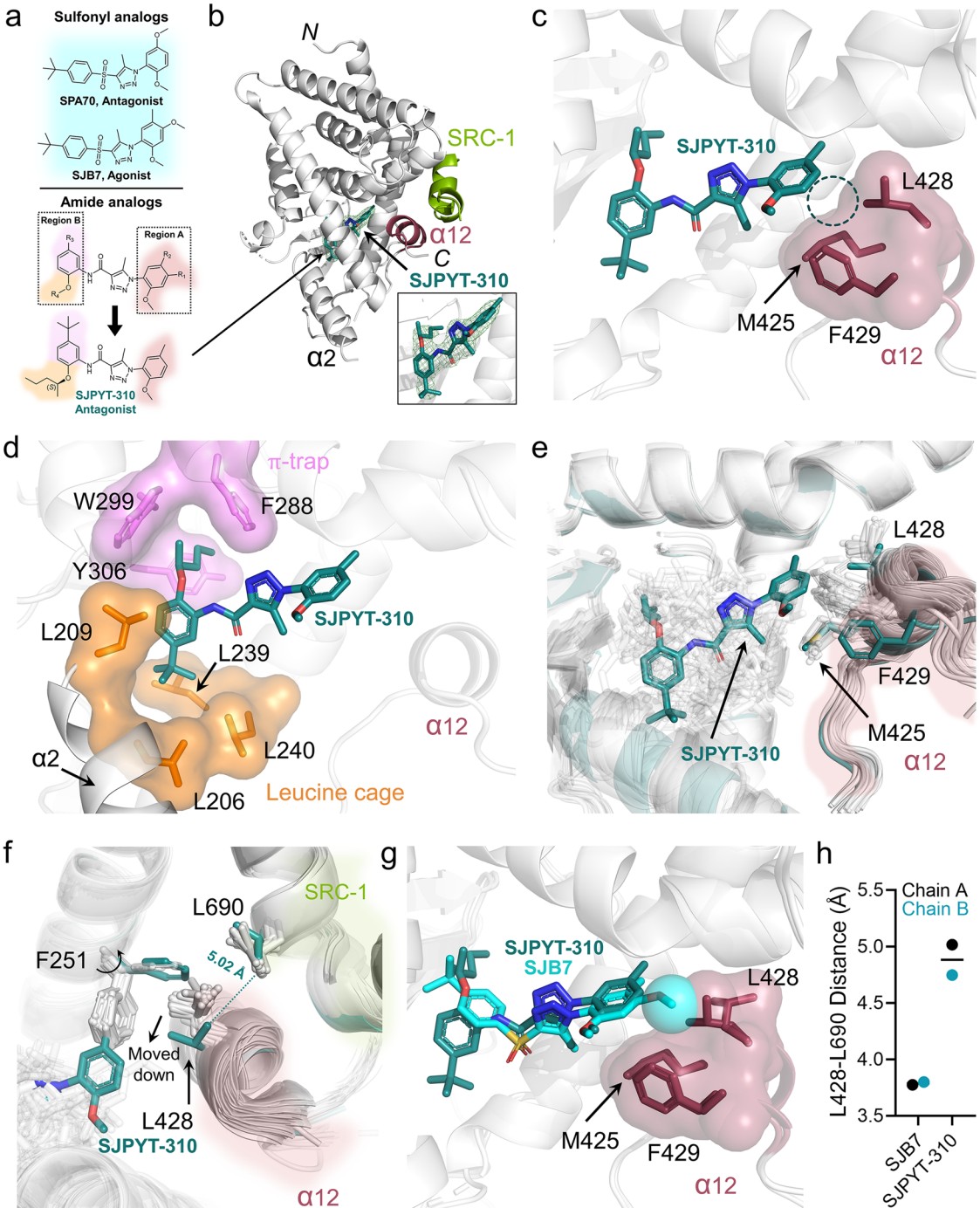

**Fig. 1 | Crystal structure of PXR LBD in complex with SJPYT-310. a** Chemical structures of representative compounds. **b** Cartoon representation of PXR LBD (gray) and tethered SRC-1 peptide (light green), with α12 shown in dark red. SJPYT-310 is clearly discernible in the *Fo-Fc* omit map (3.0 σ; green mesh). **c** SJPYT-310 is shown with ligand-facing α12 residues depicted as dark red stick representation with transparent surface model. The dashed circle indicates the missing interaction of SJPYT-310 with the α12 cleft. **d** SJPYT-310 region B interacts with π-trap residues (violet) and residues that form a "leucine cage" (orange), stabilizing α2. **e** All previously reported PXR LBD structures are overlaid. Ligands and α12 residues are shown as sticks. SJPYT-310 and the corresponding α12 residues are colored deep teal. **f** SJPYT-310 reorients L428 to lose contact with SRC-1 residue L690. **g** Overlay of SJPYT-310 and SJB7-bound PXR LBD structures (PDB ID 5XOR). The sphere indicates the 4-methoxy group of SJB7 interacting with the α12 cleft. **h** The distances between the closest atoms of PXR L428 and SRC-1 L690 were calculated for SJB7-bound and SJPYT-310-bound PXR LBD. The measurements were taken for both chains of each structure, and lines represent the mean values. Source data for (**h**) are provided as a Source Data file.

*tert*-butyl to maintain hydrophobic contacts while potentially gaining polar contacts. It should be noted that SJPYT-278 is a racemic mixture, where the (*S*) and (*R*) enantiomers were individually synthesized as SJPYT-312 and SJPYT-313, respectively. All compounds showed strong binding affinity for PXR LBD that was comparable to or greater than the potent PXR ligand T0901317 (Fig. 2b, Table 1). SJPYT-331 showed the highest binding potency, indicating that the ester substitution indeed gains favorable interactions. Agonistic and antagonistic activities were evaluated using a luciferase reporter under the control of a PXR-responsive *CYP3A4* promoter (Fig. 2c, d, Table 1)[33], and as predicted,

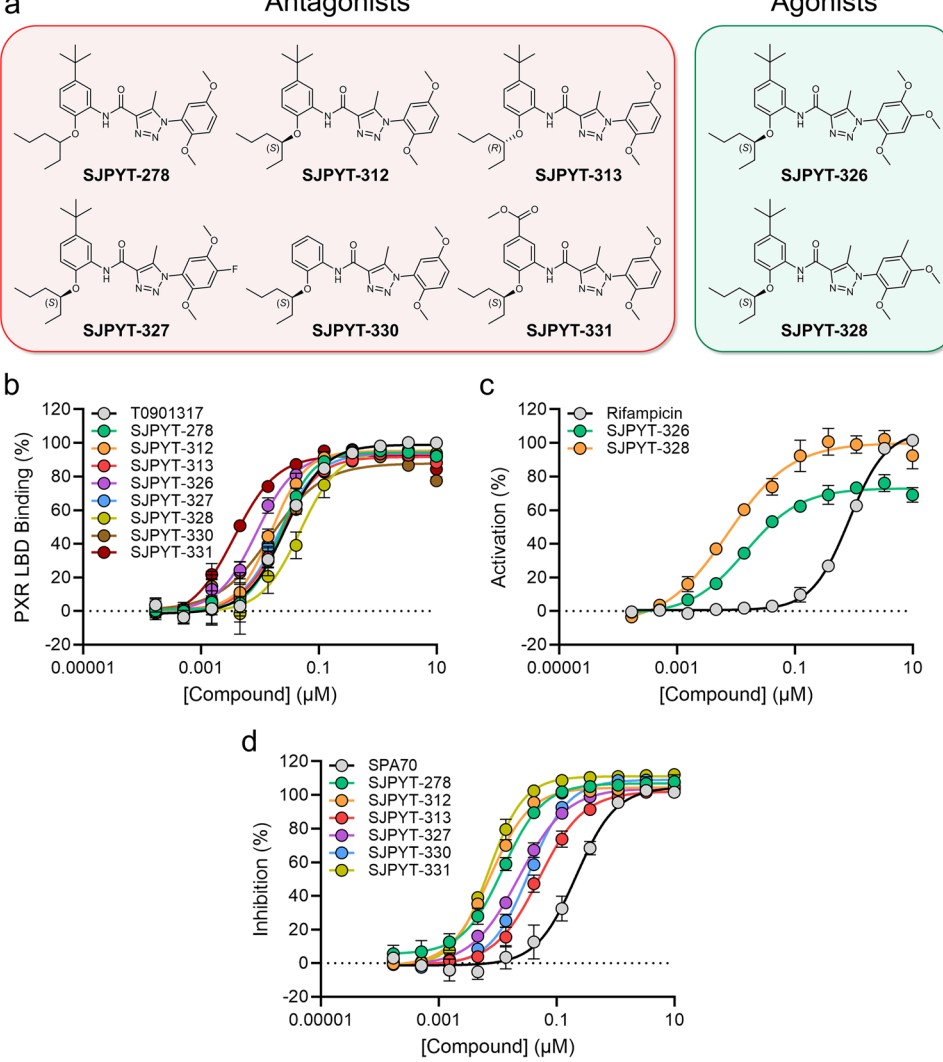

**Fig. 2 | Evaluation of PXR modulation by chemical analogs. a** Chemical structures of designed antagonists and agonists. **b** TR-FRET PXR LBD binding assay measuring displacement of a fluorescent probe from PXR LBD. Data were normalized to 10 µM T0901317 as 100% binding and DMSO as 0% binding. **c**, **d** Compounds were evaluated for PXR agonism or antagonism in HepG2 cells stably expressing PXR and a firefly luciferase reporter under the control of the PXR-responsive *CYP3A4* promoter. **c** Cellular assay in agonistic mode, where agonists increase signal. Data were normalized to 10 µM rifampicin as 100% activation and DMSO as 0% activation. **d** Cellular assay in antagonistic mode, where cells are incubated with rifampicin (5 µM), and antagonists reduce the rifampicin-induced signal. Data were normalized to 10 µM SPA70 as 100% inhibition and DMSO as 0% inhibition. Source data for (**b**–**d**) are provided as a Source Data file. Data were derived from $n = 3$ independent experiments and are presented as mean values +/- standard deviation (SD).

analogs with the 4-methoxy (SJPYT-326 and SJPYT-328) are agonists, and all compounds lacking the 4-methoxy (SJPYT-278, SJPYT-312, SJPYT-313, SJPYT-327, SJPYT-330, and SJPYT-331) are antagonists. Furthermore, SJPYT-310 and SJPYT-331 block agonist-induced PXR activation in primary human hepatocytes (PHH) (Supplementary Fig. 3a). Therefore, we were able to reliably predict PXR modulators with designated activities that translated to the gold standard liver cell system.

Because there is often ligand overlap among NRs of the NR1 family, we evaluated the compounds for modulation of farnesoid X receptor (FXR), liver X receptor α (LXRα), constitutive androstane receptor (CAR), and vitamin D receptor (VDR). We also assessed species selectivity by using mouse PXR (mPXR). The compounds were highly selective for human PXR, with no observed activation or inhibition of other receptors and no mPXR modulation in primary mouse hepatocytes (Fig. 3a, Supplementary Figs. 3b, 4, and 5). There was also no noticeable cytotoxicity in human cell models (HepG2, HEK293, and HepaRG), and mouse Hepa 1–6 cells (Supplementary Fig. 6). To

structurally evaluate the PXR selectivity, we analyzed the NR ligand binding cavities (Fig. 3b–f). The cavity volume of SJPYT-310-bound PXR LBD measures ~1421 Å$^3$, which is notably larger compared to ligand-bound FXR[34], LXRα[35], CAR[36], and VDR[37] (712 Å$^3$, 1,075 Å$^3$, 596 Å$^3$, and 757 Å$^3$, respectively). The orientation of SJPYT-310 clashes with binding pocket residues of FXR, LXRα, CAR, and VDR (Fig. 3c–f). In PXR LBD, the SJPYT-310 orientation is dictated by hydrogen bonding with Q285 (Supplementary Fig. 1a), but Q285 is not conserved among the other four NRs (Supplementary Fig. 7a). Furthermore, alignment of mPXR (model retrieved from the AlphaFold Protein Structure Database[38,39]) and SJPYT-310-bound PXR LBD shows lack of conservation of ligand-binding residues, including Q285 (I282 in mPXR) (Supplementary Fig. 7b).

## Antagonists alter AF-2 to be incompatible with coactivator binding

Based on our proposed mechanism of PXR antagonism (Fig. 1), we predicted, designed, and demonstrated that SJPYT-326 and SJPYT-328

**Table 1 | Functional and biochemical evaluation of PXR modulators**

| Compound | Binding[a] IC$_{50}$ (nM) | Agonism[b] IC$_{50}$ (nM) | Antagonism[c] IC$_{50}$ (nM) |
|---|---|---|---|
| T0901317 | 27 ± 3 | NT[d] | NT |
| Rifampicin | NT | 830 ± 40 | NT |
| SPA70 | 230 ± 30 | NA[e] | 210 ± 30 |
| SJPYT-278 | 23 ± 4 | NA | 12 ± 3 |
| SJPYT-312[f] | 15 ± 3 | NA | 7.4 ± 0.6 |
| SJPYT-313[g] | 23 ± 3 | NA | 51 ± 9 |
| SJPYT-326 | 8.7 ± 0.2 | 15 ± 0.9 | NA |
| SJPYT-327 | 21 ± 6 | NA | 23 ± 3 |
| SJPYT-328 | 48 ± 10 | 7.1 ± 0.6 | NA |
| SJPYT-330 | 14 ± 4 | NA | 34 ± 4 |
| SJPYT-331 | 3.6 ± 0.8 | NA | 7.1 ± 0.8 |

[a]PXR LBD binding assay (TR-FRET), where displacement of the fluorescence-labeled probe results in decreasing signal.
[b]Cell-based assay in agonistic mode, where agonists increase signal.
[c]Cell-based assay in antagonistic mode, where cells are incubated with rifampicin (5 µM) and antagonists reduce the rifampicin-induced signal.
[d]NT: not tested.
[e]NA: no IC$_{50}$ value could be determined experimentally within the concentration range tested.
[f]Corresponding to the (S)-enantiomer.
[g]Corresponding to the (R)-enantiomer.
All data were calculated from n = 3 independent experiments. Source data are provided as a Source Data file.

are PXR agonists, while other analogs act as antagonists (Fig. 2). To determine if the structural characteristics of these compounds are consistent with our conclusions based on SJB7 and SJPYT-310 (Fig. 1), we solved the structures of PXR LBD bound to the agonists SJPYT-326 and SJPYT-328 and the antagonists SJPYT-278, SJPYT-312, and SJPYT-331 (Fig. 4, Supplementary Figs. 8–10, Supplementary Tables 1 and 3). The electron density in PXR LBD complexed with the racemic SJPYT-278 mixture indicates the presence of both (R) and (S) enantiomers (Supplementary Figs. 8f and 10a, b), likely because they have similar PXR LBD binding potency (Fig. 2b, Table 1). The SJPYT-312 complex allowed visualization specifically of the (S) enantiomer (Fig. 4a).

As discussed above, region B interactions appear to contribute substantially to binding potency. Like SJPYT-310, region B of all analogs engages both the π-trap and the leucine cage (Fig. 4a–d), and the triazole moiety is hydrogen bonded with Q285 (Supplementary Fig. 10c–f). The binding mode of SJPYT-331 (containing the ester substitution) is largely similar to the other ligands; however, the orientation of the hexan-3-yloxy phenyl moiety is altered to bind deeper in the π-trap (Fig. 4e), likely accounting for its higher binding potency. Importantly, the region A orientations corroborate the mechanism of PXR antagonism deduced from the PXR LBD-SJPYT-310 complex. Like SJPYT-310, all compounds reposition F251 and the α12 residues L428 and F429. The antagonists SJPYT-312, SJPYT-331, and SJPYT-278 lack the region A 4-methoxy group and cannot effectively engage the α12 cleft (Figs. 4a, 4d, Supplementary Fig. 10a). The 4-methoxy groups of agonists SJPYT-326 and SJPYT-328 fill the α12 cleft, stabilizing the helix position (Fig. 4b–c). Furthermore, all antagonists consistently lose PXR L428 contact with the SRC-1 LXXLL residue L690, while the interaction is maintained by agonists (Fig. 4f, g).

Solution-based and molecular dynamics (MD) experiments have demonstrated the dynamic nature of NR α12 helices, where ligand binding fixes a subset of conformations[28,29,40–45]. We observed residue-level differences in the AF-2 region of PXR LBD bound to different ligands, but crystallographic constraints did not allow us to observe large-scale α12 movement. Molecular modeling, mutagenesis, and MD previously indicated that antagonists induce outward α12 motion[41]. Therefore, we performed MD simulations of PXR LBD bound to

agonists (SJPYT-326 and SJPYT-328) or antagonists (SJPYT-312 and SJPYT-331) (Supplementary Figs. 11–12). While α12 of agonist-bound PXR LBD remains in the crystallographically determined inward active conformation, α12 of antagonist-bound PXR LBD moves outward to a position overlapping with the SRC-1 peptide binding site (Supplementary Fig. 11a–d). Using a relatively static reference point [alpha carbon (C$_{alpha}$) of L412], we measured the movement of α12 by calculating the distance between L412 C$_{alpha}$ and F429 C$_{alpha}$ and found that the agonist complexes consistently have inward, active α12 conformations while antagonist-bound α12 is pushed outward over the course of the simulations (Supplementary Fig. 11e). Residue interaction analysis shows agonist-biased (L239, L411, and L428) and antagonist-biased (I254, H407, and F420) interactions, with L428 being the key α12 interaction for agonists (Supplementary Fig. 12). Together, the functional assays, series of PXR LBD structures, and computational analysis establish a consistent structural mechanism of antagonizing PXR by remodeling key residues in the AF-2 region. The scaffold-specific changes at the antagonist-α12 interface are unfavorable and push α12 away from the ligand binding pocket to a site incompatible with SRC-1 binding. Gain of agonistic activity through the 4-methoxy addition appears to be a "correction" of the inactivation mechanism by providing a robust hydrophobic stabilization network.

**α12 mutations convert antagonists to agonists**
Because our compound series engages the AF-2 region in unconventional manners compared to other reported PXR ligands, we next investigated the functional effects of mutating M425, L428, and F429. We and others have previously reported that M425A and F429A mutations render PXR inactive[17,46,47]. To prevent loss of function, we selected mutations that retain biochemical characteristics similar to the wild-type (WT) residues, such as hydrophobicity and size, and tested the agonist SJPYT-328 and antagonists SJPYT-310 and SJPYT-331 against M425I, M425L, L428V, L428Y, and F429I mutants in PXR-responsive reporter assays (Fig. 5a). SJPYT-328 retained agonistic behavior for all variants, indicating that the mutations are not inactive. Surprisingly, however, all five mutations converted the antagonists to agonists with varying degrees of efficacy.

To explore the ligand-mutant interactions and exclude the possibility that a different ligand orientation causes the switch from antagonist to agonist, we obtained crystal structures of PXR$^{L428V}$ LBD apo and bound to the antagonist SJPYT-331 (Fig. 5b–d, Supplementary Table 4). L428V was chosen because (1) it is potently activated by the three ligands (Fig. 5a), (2) it is a conservative mutation in which the WT residue (Leu) is biochemically similar to the mutated residue (Val), and (3) L428 shifts away from SRC-1 in the antagonist-bound WT PXR LBD structures (Fig. 4f, g). The mutation did not alter the overall protein fold, including the position of α12, as compared to WT PXR LBD (Fig. 5b, Supplementary Table 5). However, the orientation of SJPYT-331 region A twists by an angle of 14°, and the contact distances show that SJPYT-331 is closer to L428V α12 than to WT α12 (Fig. 5c–e). The L428V mutation results in increased hydrophobicity, reduced amino acid sidechain size, and reduced rotameric freedom. Accordingly, the M425-V428-F429 contact surface was slightly compacted compared to the M425-L428-F429 surface. The altered orientation of SJPYT-331 and condensed α12 surface combined with the increased hydrophobicity of Val over Leu allow the antagonist to shift toward α12 and more effectively interact through hydrophobic contacts (Fig. 5c–e), resulting in activation of the mutant. The residues M425, L428, and F429 form a hydrophobic "hot spot" that contacts agonists through moieties such as the 4-methoxy group, and mutations may change the nature of the hot spot due to a combination of factors, including hydrophobicity, orientation, and size.

## Discussion
While members of the NR superfamily share a common structural architecture, each member is uniquely regulated, contributing to

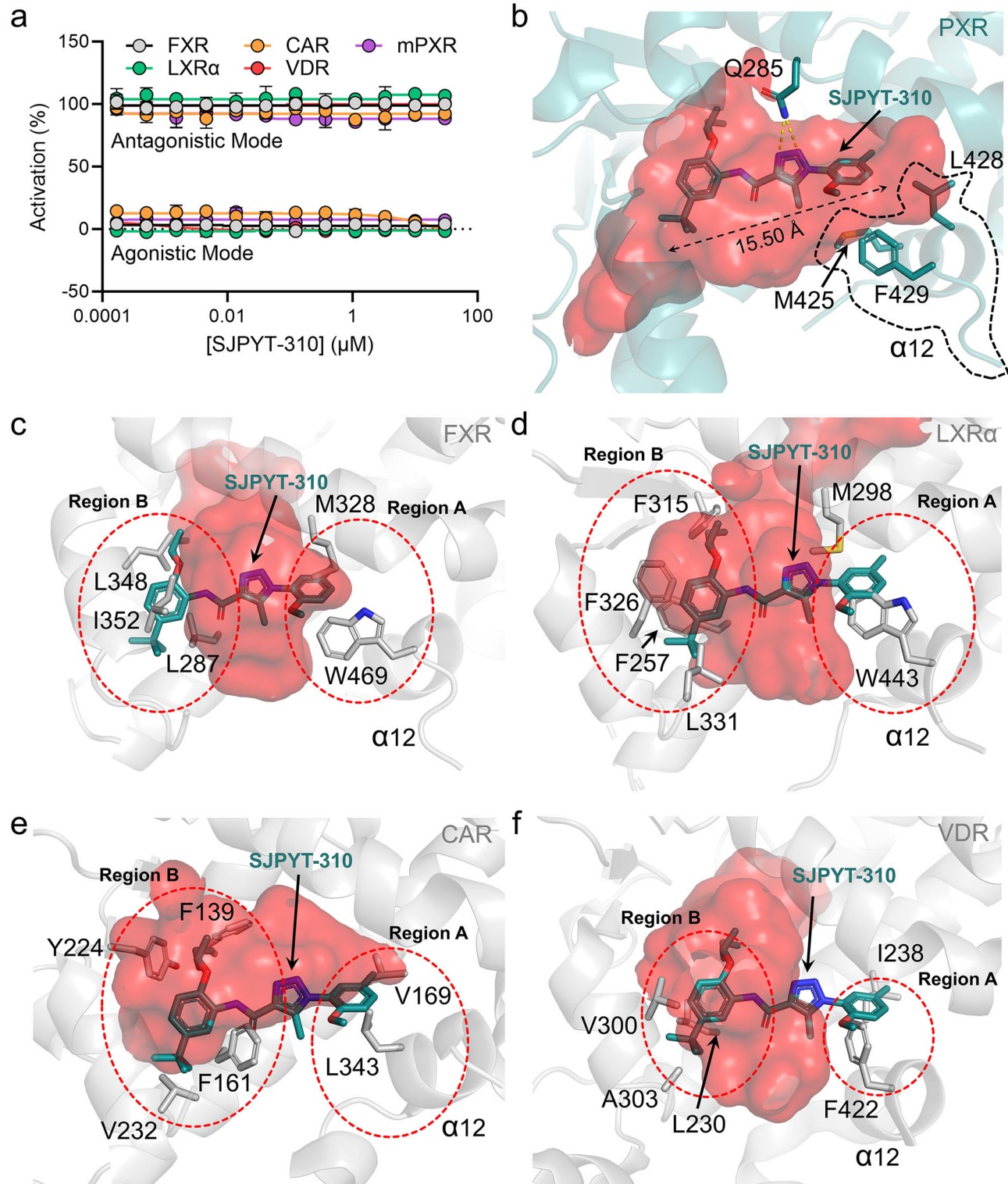

**Fig. 3 | SJPYT-310 is a PXR-selective modulator. a** SJPYT-310 was evaluated for agonism or antagonism of the indicated NRs by cellular reporter assays. In agonistic mode, cells were treated with SJPYT-310 alone. In antagonistic mode, cells were treated with SJPYT-310 in combination with NR-specific agonist (200 nM GW4064 for FXR, 300 nM T0901317 for LXRα, 300 nM CITCO for CAR, 5 nM calcitriol for VDR, and 3 μM PCN for mPXR). Data were derived from $n = 3$ independent experiments and are presented as mean values +/- SD. Source data are provided as a Source Data file. **b–f** The binding pose of SJPYT-310 in (**b**) PXR LBD is overlaid on the ligand binding pockets of (**c**) FXR (PDB ID: 6A5X), (**d**) LXRα (PDB ID: 3IPQ), (**e**) CAR (PDB ID: 1XV9), or (**f**) VDR (PDB ID: 5V39). The binding pocket cavities are shown as semi-transparent red surfaces. Residues incompatible with ligands are highlighted in red dashed circles.

diverse roles of NRs in physiological and pathological processes and many drugs developed to treat various diseases. PXR is a key player in regulating expression of drug-metabolizing enzymes and transporters that conduct the biotransformation and clearance of xenobiotics,

including therapeutic drugs[48]. This process compromises drug efficacy and can lead to the generation of toxic metabolites[49]. Unlike many other members of the NR superfamily that often have a specific ligand profile, PXR is known for its ligand promiscuity. Because of the

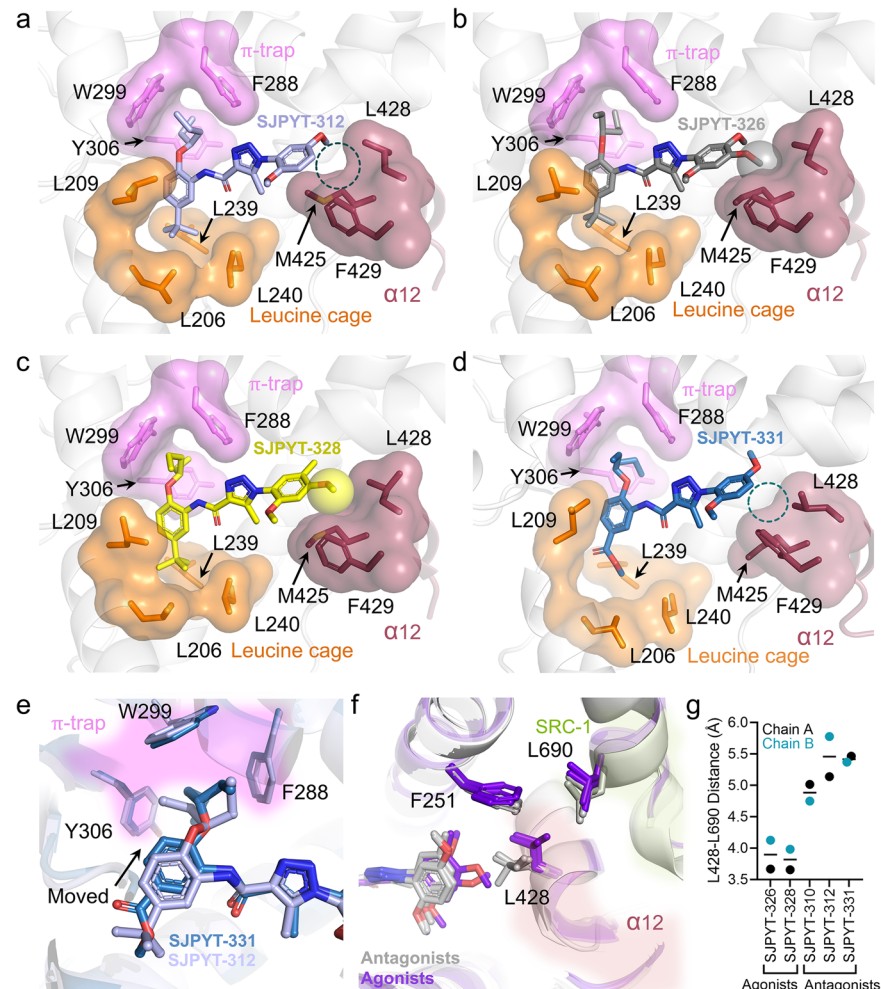

**Fig. 4 | Binding poses of the designed ligands. a–d** Crystal structures of PXR LBD complexed with SJPYT-312, SJPYT-326, SJPYT-328, or SJPYT-331. Region B moieties bind to the π-trap (violet) and leucine cage (orange), and region A groups are oriented toward α12 (dark red). The 4-methoxy groups (region A) of agonists SJPYT-326 and SJPYT-328 interact with the α12 cleft (spheres), and this interaction is absent in the antagonists SJPYT-312 and SJPYT-331 (dashed circles). **e** Overlay of PXR LBD bound to SJPYT-312 or SJPYT-331. SJPYT-331 binds tightly to the π-trap.

**f** Overlay of PXR LBD bound to SJPYT-310, SJPYT-312, SJPYT-331, SJPYT-326, SJPYT-328, or SJB7. All antagonist structures are shown in gray, and all agonist structures are shown in purple. When antagonist is bound, PXR L428 shifts away from SRC-1 L690. **g** The distances between the closest atoms of PXR L428 and SRC-1 L690 were calculated for the indicated complexes. The measurements were taken for both chains of each structure, and lines represent the mean values. Source data for (**g**) are provided as a Source Data file.

propensity of many medications, such as commonly prescribed drugs like rifampicin and paclitaxel, to bind and activate PXR, PXR antagonists may be co-administered to pharmacologically prevent drug toxicity and drug resistance. However, development of antagonists has faced imposing challenges due to PXR's ligand binding pocket being adapted to recognize structurally diverse agonists[48,50]. The promiscuous nature of PXR is attributed to its LBD, which is larger and more dynamic than the LBDs of other NRs[51,52]. It can expand its volume substantially, enabling it to accommodate large ligands or even multiple ligands simultaneously[14,15,21,27]. Although most PXR ligands are agonists, a few compounds have been identified that inhibit PXR's transcriptional activation[53]. However, the molecular basis of PXR antagonism has remained elusive due to insufficient biochemical characterization and absence of structural studies. Here, we present a structural mechanism by which PXR activity can be inhibited or activated by closely related analogs with small chemical changes.

To facilitate protein stability during purification and crystallization, inclusion of a small fragment of SRC-1 has been widely employed, either as a free peptide or tethered to the C-terminus of PXR LBD[31,54]. Moreover, the size and flexibility of the PXR ligand binding

pocket allow ligands to adopt several conformations[30], and the SRC-1 peptide has been shown to restrict ligand orientations[54]. In all reported PXR LBD structures, there is no difference in α12 conformation regardless of the presence or absence of ligand[55]; indeed, most NR structures display the same α12 conformation[50]. The observation that ligand binding does not affect α12 conformation in crystal studies provided us with the opportunity to determine differences in interactions between agonists and antagonists to residues of the PXR ligand binding pocket under similar structural circumstances. Furthermore, the use of SRC-1 peptide allowed us to observe antagonist-induced changes at the AF-2/SRC-1 interface that have not been visualized in previous PXR LBD studies. Specifically, antagonist binding resulted in a loss of PXR α12 L428 contact with L690 of the SRC-1 LXXLL motif (Fig. 4f, g). Because the LXXLL motif is a requirement for NR interaction[26,54], this loss of contact may represent a means by which antagonists either prevent SRC-1 engagement or promote SRC-1 disengagement.

α12 is the dominant determinant in dictating ligand-mediated NR activation[28,29]. Agonists affix α12 in an orientation amenable for the recruitment of a coactivator. In-solution studies indicate that in the absence of ligand, α12 is mobile, and mobility is reduced upon binding

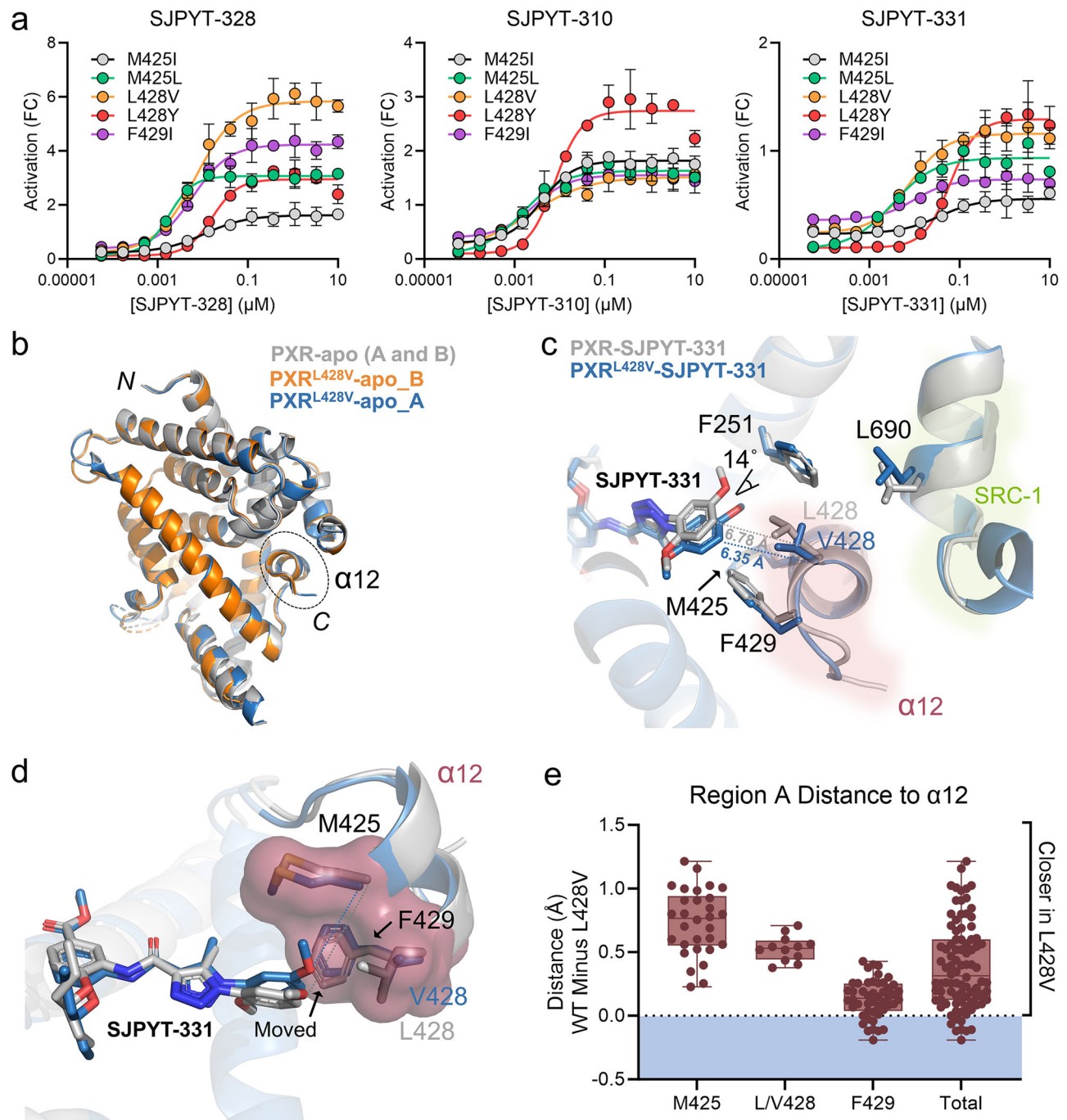

**Fig. 5 | Mutations of the α12 residues convert antagonists to agonists. a** HepG2 cells were transfected with a PXR-expressing plasmid and a plasmid encoding firefly luciferase under the control of a PXR-responsive *CYP3A4* promoter. Cells were treated with agonist SJPYT-328, antagonist SJPYT-310, or antagonist SJPYT-331. Results are normalized as fold change (FC) relative to HepG2 cells transfected with WT PXR and treated with DMSO. Data were derived from $n = 3$ independent experiments and are presented as mean values +/- SD. **b** The crystal structure of apo PXR^L428V LBD was solved, showing similar overall structure as WT PXR LBD. **c**, **d** The crystal structure of SJPYT-331-bound PXR^L428V LBD was solved. In PXR^L428V LBD,

SJPYT-331 shifted with an angle of 14˚ towards the key α12 residues compared to WT PXR LBD. Distances are shown as dashed lines, indicating that SJPYT-331 in PXR^L428V is closer to α12 residues. **e** All-atom pairwise distances between the region A benzene ring and the sidechains of M425, L/V428, and F429 were calculated for WT and L428V PXR LBD (only the alpha and beta carbons were used for the L/V428 measurement). Each point represents the WT distance minus the paired L428V distance (e.g., a value of 1.0 indicates the SJPYT-331 atom is 1.0 Å closer to the α12 atom in L428V compared to WT). Source data for (**a**) and (**e**) are provided as a Source Data file.

of an agonist to the NR LBD[45,56]. Two major classes of NR antagonists have been broadly identified based on the manner in which the ligand prevents α12 from orienting in the active conformation: those that physically obstruct α12 from positioning into the active form and those that fail to stabilize the mobile α12 favorably due to inadequate interactions with key residues of the helix[28,50,57]. The PXR antagonists

presented in our work appear to follow the latter mechanism, where ligands lacking the region A 4-methoxy group have insufficient contacts with the α12 hydrophobic cleft formed by M425, L428, and F429. This is supported by the observation that simple addition of the 4-methoxy restores agonist activity. Furthermore, MD simulations showed that agonists maintain the active α12 conformation while

antagonists push α12 into space that would be occupied by SRC-1 (Supplementary Fig. 11).

In summary, we present a structural mechanism of PXR antagonism, where a specific chemical scaffold rearranges the AF-2/SRC-1 interface. Using structure-guided design, we could predictably manipulate the agonistic and antagonistic properties of a series of analogs through minor chemical modifications. Notably, these ligands display high potency with no noticeable toxicity and are selective for PXR over a panel of related NRs, although these properties are yet to be evaluated in vivo. Our findings open possibilities for the development of improved PXR antagonists that can be applied in a clinical setting as cotreatment with existing drugs and provide a chemical framework for modifying drugs to avoid their activation of PXR.

## Methods

### Cell culture and reagents

Key reagents used in this study are summarized in Supplementary Data 1. Human hepatocellular carcinoma HepG2/C3A (henceforth referred to as HepG2, cat. # CRL-3581), mouse hepatoma Hepa 1–6 (cat. # CRL-1830), and HEK293 (cat. # CRL-1573) cells were obtained from the American Type Culture Collection (ATCC, VA, USA). GeneBLAzer NR-UAS-*bla* HEK 293 T assays for VDR (cat. # K1099), FXR (cat. # K1691B), and LXRα (cat. # K1692) were obtained from Invitrogen (now Thermo Fisher Scientific, CA, USA) and maintained according to the manufacturer's instructions. Human hepatoma HepaRG cells (cat. # HPRGC1, lot # 1116308B) were purchased from Thermo Fisher Scientific and maintained in Williams' Medium E (Sigma-Aldrich, MO, USA, cat. # W1878-500ml) supplemented with HepaRG Thaw, Plate, and General Purpose Medium Supplement (Thermo Fisher Scientific, Gibco cat. # HPRG670) as previously described[58]. HepG2 cells were maintained in Eagle's Minimum Essential Medium (EMEM, ATCC, cat. # 30-2003) with 10% fetal bovine serum (FBS) (HyClone, Danaher Corp., D.C., USA). HEK293 cells were cultured in Dulbecco's Modified Eagle Medium (DMEM, ATCC, cat. # 30-2002) with 10% FBS. Hepa 1–6 cells were cultured in DMEM (ATCC, cat. # 30-2002) with 10% FBS. Primary Human Hepatocytes (PHH), Cryopreserved, Plateable and Interaction Qualified were purchased from Lonza (Basel, Switzerland, cat. # HUCPI). Lot numbers for donors 1–3 were HUM190171, HUM211621, and HUM200271, respectively. Mouse CD-1 Hepatocytes, Cryopreserved, Plateable were purchased from Lonza (cat. # MCCP01/ROW, lot # MCD203). PHH and mouse hepatocytes were thawed and cultured as described in the specific subsection below. All cell lines were incubated in a humidified atmosphere at 37 °C with 5% CO₂, were authenticated by short tandem repeat (STR) DNA profiling, and were routinely verified to be mycoplasma free by using the MycoProbe Mycoplasma Detection Kit (R&D Systems, Inc., MN, USA). Cell counts were obtained with a Countess II Automated Cell Counter (Thermo Fisher Scientific) using trypan blue staining. Phenol red–free DMEM, Tb-anti-GST (terbium-antiglutathione S-transferase, cat. # PV3550), GST-PXR-LBD (cat. # PV4841), Tris (pH 7.5, 1 M), and dithiothreitol (DTT, 1 M) were purchased from Thermo Fisher Scientific. MgCl₂ (1 M) was purchased from Boston BioProducts (MA, USA). Rifampicin, CITCO, 1α,25-dihydroxyvitamin D3 (calcitriol), GW4064, and bovine serum albumin (BSA) were purchased from Sigma-Aldrich. T0901317 and PCN were purchased from Cayman Chemical (MI, USA). Staurosporine was purchased from LC Laboratories (MA, USA). SPA70 was prepared as previously reported[17]. Gibco Geneticin (G418) was purchased from Thermo Fisher Scientific (cat. #10-131-027). Steadylite Plus Reporter Gene Assay System and 384-well white tissue culture-treated plates were purchased from PerkinElmer Life Sciences (MA, USA). Black 384-well low-volume assay plates and collagen (cat. # 354236) were purchased from Corning Inc. (MA, USA). BODIPY FL vindoline was synthesized in house as previously reported[59]. CellTiter-Glo Luminescent Cell Viability Assay reagent and Dual-Glo luciferase assay were purchased from Promega (WI, USA). Charcoal/dextran-treated FBS was purchased from HyClone Laboratories.

### Plasmids

Construction of the mammalian pcDNA3-FLAG-PXR expression plasmid was described previously using pcDNA3 from Thermo Fisher Scientific[33]. Mutations were made using the Q5 Site-Directed Mutagenesis Kit and primers listed in Supplementary Data 1. Construction of pGL3-*CYP3A4*-luc containing firefly luciferase under the control of a PXR-responsive *CYP3A4* promoter has also been described[33,60]. pCMV6-mPXR was purchased from OriGene Technologies, MD, USA, (cat. # MR226044). pRL-TK was purchased from Promega. Construction of the bacterial pET-3a PXR LBD and PXR[L428V] LBD expression plasmids was described below.

### Protein expression and purification

We used a plasmid expressing PXR LBD tethered to a 33-amino acid SRC-1 peptide, as previously described[21]. A codon-optimized sequence for human PXR LBD (residues 130-434) fused to SRC-1 (residues 678-710) with SGGSGG linker was cloned into bacterial expression pET-3a vector with a His6-tag at the N-terminus (Novagen, Merck group, Darmstadt, Germany). The L428V mutant was generated using the Q5 Site-Directed Mutagenesis Kit (New England Biolabs, MA, USA) and primers listed in Supplementary Data 1. The plasmid was transformed into TurboCells Competent *E. coli* BL21(DE3) (Genlantis, CA, USA), grown in terrific broth (Legacy Biologicals, IL) supplemented with 0.2% glucose (Sigma-Aldrich) at 37 °C to an OD₆₀₀ of 3-4 and induced overnight at 16 °C with 200 μM IPTG (Gold Biotechnology, MO, USA). Cells were pelleted by centrifugation at 4000 × g and resuspended in lysis buffer [25 mM Tris (Research Products International, IL, USA) (pH 7.9), 500 mM NaCl (Research Products International), 5% (v/v) glycerol (Thermo Fisher Scientific), 1 mM DTT (Gold Biotechnology), 10 mM imidazole (Thermo Fisher Scientific)] supplemented with EDTA-free SIGMAFAST protease inhibitor cocktail tablets (Sigma-Aldrich). The suspension was lysed using sonication and centrifuged at 20,000 × g for 1 h, and the supernatant was applied to a 5 mL HisTrap FF column (Cytiva, Danaher Corp., D.C., USA). The column was washed with 50 mL lysis buffer, and bound proteins were eluted with a 100 mL linear gradient from lysis buffer to lysis buffer containing 250 mM imidazole. Elution fractions were collected and analyzed by SDS-PAGE for protein amount and purity. Selected fractions were pooled and diluted to 125 mM NaCl by addition of lysis buffer without NaCl. The protein was applied to a 5 mL HiTrap SP HP column (Cytiva, Danaher Corp.), and the flow-through contained the PXR LBD or PXR[L428V] LBD. The protein was concentrated to ≤ 10 mL in an Amicon Ultra-15 centrifugal filter unit (Millipore, Darmstadt, Germany) with 10 kDa cutoff, filtered through a 0.22 μm syringe filter, and loaded onto a HiLoad 26/600 Superdex 200 pg size exclusion column (Cytiva, Danaher Corp.) equilibrated with storage buffer [25 mM Tris (pH 8.0), 100 mM NaCl, 5% (v/v) glycerol, 2 mM DTT]. Elution fractions were collected and analyzed by SDS-PAGE, and pure fractions were pooled, concentrated to 12 mg/mL, aliquoted, flash frozen in liquid nitrogen, and stored at -80 °C.

### Crystal structure determination of PXR LBD complexed with ligands

Previously reported crystallization conditions were chosen for optimization[21]. PXR LBD or PXR[L428V] LBD (12 mg/mL) with 2:1 molar ratio of ligand (SJPYT-278, SJPYT-310, SJPYT-312, SJPYT-326, SJPYT-328, or SJPYT-331) were incubated for 1 h at 4 °C. The mixture contained ~2% DMSO from the compound dilution. Apo crystals were obtained by using the same protein concentration. The mixtures were set up for crystallization in 24-well VDX hanging drop plates (Hampton Research Corp., CA, USA) with drops consisting of equal volumes of protein and reservoir solutions. Crystals formed for apo PXR LBD and grew over

8–10 days in 0.1 M HEPES pH 7.0, and 14% isopropanol (Sigma-Aldrich) at 4 °C. Apo PXR$^{L428V}$ LBD or co-crystals of all complexes were obtained in 50 mM Bis-Tris (pH 6-7), and 10-20% (+/−)-2-methyl-2,4-pentanediol (MPD) (Hampton Research Corp.) and grew within 24 h at 20 °C. Crystals were cryoprotected in reservoir solution supplemented with 25-30% (v/v) MPD and flash-frozen in liquid nitrogen. All X-ray diffraction data were collected from single crystals. Data for apo PXR LBD, PXR LBD complexes with SJPYT-310, SJPYT-312, SJPYT-326, SJPYT-328, and PXR$^{L428V}$ LBD with SJPYT-331 were collected to resolutions of 2.20, 2.35, 2.75, 2.92, 2.68, and 2.14 Å, respectively, at SER-CAT Beamline 22-ID at the Advanced Photon Source at the Argonne National Laboratory. Data for apo PXR$^{L428V}$ LBD, and PXR LBD complexes with SJPYT-278 and SJPYT-331 were collected to resolutions of 2.89, 3.32 and 2.39 Å, respectively, at AMX and FMX Beamlines (17-ID-1 and 17-ID-2) at the National Synchrotron Light Source II at Brookhaven National Laboratory. Frames were processed with XDS[61], and all crystals belonged to space group P2$_1$2$_1$2$_1$ with two molecules in the asymmetric unit. The structures were solved by molecular replacement in Phaser[62] using PDB ID 3CTB as the search model[31]. The search model was stripped of solvent prior to molecular replacement. Iterative cycles of model building, and refinement were performed in Coot[63] and Phenix[64]. To visualize unbiased densities for ligands, omit maps were generated by omitting the ligands using the phenix.polder program[64]. The 2Fo-Fc maps of each ligand in the asymmetric unit and 2D ligand interaction diagrams are shown in Supplementary Figs. 8–9, respectively All crystallographic figures were made in PyMOL (Schrödinger). The calculations of cavities volume were performed using CASTp[65] with a sphere radius 1.4 Å. Data processing and refinement statistics are shown in Supplementary Tables 1, 3, and 4. Distance measurements were performed with pairwisedistances.py (https://pymolwiki.org/index.php/Pairwise_distances). 2D ligand interaction diagrams were generated using the Molecular Operating Environment (MOE, v.2022.02, Chemical Computing Group ULC).

## MD simulations

Protein preparation and simulations were carried out similarly as previously described, with minor modifications[41]. Structures of PXR LBD bound to SJPYT-312 (chain B), SJPYT-326 (chain B), SJPYT-328 (chain A), or SJPYT-331 (chain A) were loaded into Coot, and missing loops were filled manually using the AlphaFold model of PXR as a guide[38,39]. The structures were then loaded into Maestro software (Schrödinger Release 2023-2), and the Protein Preparation Workflow was used to assign bond orders, add hydrogens at pH 7.4, and fill in missing side chains. Default parameters were used for optimization of hydrogen bond assignment. The Desmond System Builder was used to solvate an orthorhombic box with the SPC solvent model and neutralizing counterions with a 10 Å distance between the protein and box edge, and the forcefield was set to OPLS4[66]. Simulations were conducted in Desmond at 300 K and 1 atm for 200 ns, with 200 ps trajectory recording intervals. The system energy was set to the default value of 1.2, the ensemble class was NPT, the Nosé–Hoover thermostat was used with 1 ps relaxation time (tau), the Martyna-Tobias-Klein barostat was used with *tau* = 2 ps, and the non-bonded cutoff was 9 Å. The default option to relax systems before simulations was selected, which equilibrated the system with (1) 100 ps NVT ensemble using Brownian dynamics with restrained non-hydrogen atoms at 10 K, (2) 12 ps NVT ensemble using Langevin thermostat (*tau* = 0.1 ps) and restrained non-hydrogen atoms at 10 K, (3) 12 ps NPT ensemble using Langevin thermostat (*tau* = 0.1 ps) and barostat (*tau* = 50 ps) and restrained non-hydrogen atoms at 10 K and 1 atm, (4) 12 ps NPT ensemble using Langevin thermostat (*tau* = 0.1 ps) and barostat (*tau* = 50 ps), and restrained non-hydrogen atoms at 300 K and 1 atm, and (5) 24 ps NPT ensemble using Langevin thermostat (*tau* = 0.1 ps) and barostat (*tau* = 2 ps) without restraints at 300 K and 1 atm. Coordinates of MD inputs and outputs and representative setup files are provided as Supplementary Data 2.

## TR-FRET PXR competitive binding assays

The TR-FRET binding assays using BODIPY FL vindoline were performed similarly as previously described[24]. Experiments were conducted in a black 384-well low-volume assay plate, with 20 μL reaction mixture per well consisting of GST-PXR-LBD (5 nM), Tb-anti-GST (5 nM, corresponding to a 1:680 dilution of the 3.4 μM stock), BODIPY FL vindoline (100 nM), and assay buffer [50 mM Tris (pH 7.5), 20 mM MgCl$_2$, 0.1 mg/mL BSA, and 0.05 mM DTT]. The reaction mixture was incubated with test compounds, negative control (DMSO), or positive control (10 μM T0901317) for 30 min. A PHERAstar FS plate reader (BMG LABTECH, NC, USA) was used to detect the TR-FRET signals, expressed as 520 nm/490 nm. The activity of test compounds was normalized to the positive (10 μM T0901317) and negative control (0.3% DMSO).

## Nuclear receptor transactivation assays

The PXR transactivation assays (agonistic and antagonistic modes) were performed using HepG2 cells stably expressing FLAG-PXR and *CYP3A4*-luciferase reporter as previously described[24]. Test compounds, either alone (agonistic mode) or in combination with 5 μM rifampicin (antagonistic mode), were added to the wells of white 384-well tissue culture-treated plates with 5000 cells in 25 μL of phenol red-free DMEM supplemented with 5% charcoal/dextran-treated FBS and incubated for 24 h at 37 °C prior to the addition of Steadylite Plus Reporter Gene Assay System. The luminescence signal was detected with an EnVision plate reader (PerkinElmer Life Sciences). In the agonistic mode, data were normalized to rifampicin (10 μM) as 100% activation and DMSO (0.3%) as 0% activation. In the antagonistic mode, data were normalized to SPA70 (10 μM) as 100% inhibition and DMSO (0.3%) as 0% inhibition.

For the evaluation of test compounds against a panel of nuclear receptors (GeneBLAzer NR-UAS-*bla* HEK 293 T assays for VDR, FXR, or LXRα), the LiveBLAzer FRET-B/G Loading Kit with CCF4-AM, and all tissue culture reagents were purchased from Thermo Fisher Scientific and performed as previously described[24]. Test compounds alone (agonistic mode) or in combination with a specific agonist [antagonistic mode; 5 nM 1α,25- dihydroxyvitamin D3 (calcitriol) for VDR, 200 nM GW4064 for FXR, 300 nM T0901317 for LXRα] were incubated with an optimized number of cells for each assay in wells of black 384-well tissue culture-treated clear-bottom plates with 30 μL of the respective assay medium in a 37 °C cell culture incubator. Wells with no cells (containing assay medium and DMSO) were used as background control. After 24 h, 6 μL per well of loading solution (included in the kit) was added. Following incubation at room temperature in the dark (incubation time was optimized for each assay) the fluorescent emissions at 460 nm and 535 nm (using excitation at 400 nm) were used to determine the 460 nm/535 nm ratio using an EnVision plate reader. The background subtracted signal was used for analysis. In agonistic mode, the activity of the reference agonist used for each receptor [1α,25- dihydroxyvitamin D3 (calcitriol, 100 nM for VDR), 10 μM GW4064 for FXR, and 2 μM T0901317 for LXRα] was set as 100% activation and DMSO (0.3%) as 0% activation. In antagonistic mode, the activity of 5 nM 1α,25- dihydroxyvitamin D3 (calcitriol), 200 nM GW4064, and 300 nM T0901317 was set as 100% activation for VDR, FXR, and LXRα, respectively.

The CAR transactivation assays (agonistic and antagonistic modes) were performed using HepG2 cells stably expressing FLAG-CAR and *CYP2B6*-luciferase reporter[24]. The stable cells were maintained in EMEM supplemented with 10% FBS and 0.5 mg/mL G418. 5000 cells in 25 μL of phenol red-free DMEM supplemented with 5% charcoal/dextran-treated FBS were seeded into each well of 384-well white culture plates and treated with test compounds for an additional

24 h before a luciferase assay was performed using the Steadylite Plus Reporter Gene Assay System. Cells were treated with test compounds alone (agonistic mode) or in combination with the agonist CITCO (300 nM, antagonistic mode). The final DMSO concentration was 0.3% in all assays. In agonistic mode, 1 μM CITCO was used as the reference agonist for 100% activation, while DMSO was used as negative control for 0% activation. In antagonistic mode, the activity of 300 nM CITCO was set as 100% activation.

For the mPXR transactivation assay, Hepa 1–6 cells were co-transfected with pCMV6-mPXR, pGL3-*CYP3A4*-luc and pRL-TK by using Lipofectamine 3000 (Thermo Fisher Scientific). After 24 h, 5000 cells per well were treated with test compounds with or without 3 μM PCN in 384-well white culture plates (PerkinElmer Life Sciences) for 24 h in phenol red-free DMEM supplemented with 5% charcoal/dextran-treated FBS before performing the Dual-Glo luciferase assay. The *Renilla* luciferase activity was used as a reference (transfection control). The final DMSO concentration was 0.3% in all assay wells. In agonistic mode, 10 μM PCN was used as the reference agonist for 100% activation, while DMSO was used as negative control for 0% activation. In antagonistic mode, the activity of 3 μM PCN was set as 100% activation.

Transactivation assays with PXR mutants were performed similarly as previously described[17,67]. HepG2 cells (750,000/well) were plated in six-well tissue culture-treated plates. The following day, cells were co-transfected with pGL3-*CYP3A4*-luc (2 μg/well) and 100 ng/well of either empty vector (pcDNA3) or pcDNA3-FLAG-PXR (WT or mutant) using Lipofectamine 3000 (Thermo Fisher Scientific). Twenty-four hours after transfection, cells were trypsinized and suspended in phenol red-free DMEM supplemented with 5% charcoal/dextran-treated FBS, and 10,000 cells/well were added to white 384-well plates. An Echo 655 Acoustic Liquid Handler (Labcyte Inc., CA, USA) was used to dispense 75 nL/well compound stocks or DMSO, resulting in 0.3% final DMSO concentration and the indicated concentrations of test compounds. After 24 h, a luciferase assay was performed using the Steadylite Plus Reporter Gene Assay System and EnVision microplate reader (PerkinElmer Life Sciences).

## Cytotoxicity

For cytotoxicity assays, cells, test compounds, staurosporine (56 μM, positive control), or DMSO (0.5%, negative control) were added to the wells of white 384-well tissue culture-treated plates containing HepG2 (5000 cells for 24 h cytotoxicity or 2500 cells for 72 h cytotoxicity), HepaRG (5000 cells for 24 h cytotoxicity or 2500 cells for 72 h cytotoxicity), HEK293 (20,000 cells for 24 h cytotoxicity or 2500 cells for 72 h cytotoxicity), or Hepa 1–6 (5000 cells for 24 h cytotoxicity or 2500 cells for 72 h cytotoxicity) in 25 μL of phenol red-free DMEM supplemented with 5% charcoal/dextran-treated FBS (for the 24 h assay) or DMEM supplemented with 10% FBS (for the 72 h assay). After 24 or 72 h at 37 °C, the cell viability was determined using a CellTiter-Glo luminescent cell viability assay (Promega) by measuring the luminescence signal in an EnVision plate reader. The final DMSO concentration was 0.5% for all wells.

## Primary human hepatocyte (PHH) and primary mouse hepatocyte induction

Thawing and plating of cryopreserved PHH were performed according to the manufacturer's recommendations. Cells were thawed at 37 °C, transferred to Human Cryopreserved Hepatocyte Thawing Medium (Lonza), and centrifuged for 8 min at 100 × g at room temperature. The supernatant was aspirated, and cells were resuspended in Hepatocyte Plating Medium (Lonza, cat. # MP100) at 1 × 10^6 viable cells/mL and added to collagen coated 24-well plates (500 μL/well). The medium was replaced with fresh Hepatocyte Plating Medium after 1 h at 37 °C and with Hepatocyte Culture Medium (Lonza) after an additional 5 h. The following day, the medium was replaced with fresh Hepatocyte

Culture Medium containing 0.2% DMSO and the indicated compounds. After 24 h, the medium was removed, cells were washed with Dulbecco's Phosphate Buffered Saline (DPBS, Thermo Fisher Scientific), RNA was extracted with Maxwell 16 LEV SimplyRNA Tissue Kits (Promega), and cDNA was generated from 250 ng of RNA with the SuperScript VILO cDNA Synthesis Kit (Thermo Fisher Scientific). Quantitative real-time polymerase chain reaction (RT-qPCR) was conducted with 2 μL of cDNA using TaqMan Fast Advanced Master Mix (Applied Biosystems, CA, USA) in an Applied Biosystems 7500 Fast Real-Time PCR System. TaqMan gene expression assays specific for *CYP3A4* (cat. # Hs00604506_m1) and *RNA18S* (cat. # Hs03928990_g1) were purchased from Thermo Fisher Scientific. Fold induction values were calculated according to the $2^{-\Delta\Delta Ct}$ method, where $\Delta Ct$ represents the differences in cycle threshold numbers between the target gene (*CYP3A4*) and reference gene (*RNA18S*), and $\Delta\Delta Ct$ represents the relative change in these differences between the control (DMSO) and treatment (test compound) groups[68].

Thawing and plating of cryopreserved mouse hepatocytes were performed according to the manufacturer's recommendations. Cells were thawed at 37 °C, transferred to Rodent Cryopreserved Hepatocyte Thawing Medium (Lonza, cat. # MCRT50), and centrifuged for 4 min at 100 × g at room temperature. The supernatant was aspirated, and cells were resuspended in Hepatocyte Plating Medium at 0.5 × 10^6 viable cells/mL and added to collagen coated 12-well plates (1 mL/well). The medium was replaced with Hepatocyte Maintenance Medium (Lonza, cat. # MM250) after 6 h. The following day, the medium was replaced with fresh Hepatocyte Maintenance Medium containing 0.2% DMSO and the indicated compounds. After 48 h, cells were harvested and processed as above. TaqMan gene expression assays specific for *Cyp3a11* (cat. # Mm00731567_m1) and *Rn18s* (Mm03928990_g1) (Thermo Fisher Scientific) were used for RT-qPCR.

## Compound synthesis

Detailed synthetic procedures and characterization of the generated compounds can be found in the Supplementary Methods and Supplementary Figs. 13–51.

## Reporting summary

Further information on research design is available in the Nature Portfolio Reporting Summary linked to this article.

# Data availability

Source data for relevant figures are provided with this paper. PDB files and structure factors have been deposited in the RCSB PDB under the codes: 8SVN; Crystal structure of the apo form of pregnane X receptor ligand binding domain [https://doi.org/10.2210/pdb8SVN/pdb]. 8SVO; Crystal structure of pregnane X receptor ligand binding domain in complex with SJPYT-310 [https://doi.org/10.2210/pdb8SVO/pdb]. 8SVP; Crystal structure of pregnane X receptor ligand binding domain in complex with SJPYT-278 [https://doi.org/10.2210/pdb8SVP/pdb]. 8SVQ; Crystal structure of pregnane X receptor ligand binding domain in complex with SJPYT-312 [https://doi.org/10.2210/pdb8SVQ/pdb]. 8SVR; Crystal structure of pregnane X receptor ligand binding domain in complex with SJPYT-326 [https://doi.org/10.2210/pdb8SVR/pdb]. 8SVS; Crystal structure of pregnane X receptor ligand binding domain in complex with SJPYT-328 [https://doi.org/10.2210/pdb8SVS/pdb]. 8SVT; Crystal structure of pregnane X receptor ligand binding domain in complex with SJPYT-331 [https://doi.org/10.2210/pdb8SVT/pdb]. 8SVU; Crystal structure of the L428V mutant of pregnane X receptor ligand binding domain in apo form [https://doi.org/10.2210/pdb8SVU/pdb]. 8SVX; Crystal structure of the L428V mutant of pregnane X receptor ligand binding domain in complex with SJPYT-331 [https://doi.org/10.2210/pdb8SVX/pdb]. Additional previously published PDB depositions specifically mentioned are: 5X0R, 3CTB, 6A5X, 3IPQ, 1XV9, and 5V39. Source data are provided with this paper.

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

## Acknowledgements

Research reported in this publication was supported by the National Institute of General Medical Sciences [Grant R35GM118041, awarded to TC]. The content is solely the responsibility of the authors and does not necessarily represent the official views of the National Institutes of Health. Crystallographic data were collected at Southeast Regional Collaborative Access Team (SER-CAT) 22-ID beamline at the Advanced Photon Source, Argonne National Laboratory, Chicago and AMX Beamline (17-ID-1) at the National Synchrotron Light Source II, Brookhaven National Laboratory, New York. SER-CAT is supported by its member institutions, and equipment grants (S10_RR25528, S10_RR028976 and S10_OD027000) from the National Institutes of Health. The AMX beamline at the National Synchrotron Light Source II, Brookhaven National Laboratory, New York, is primarily supported by the NIH National Institute of General Medical Sciences through a Center Core P30 Grant (P30GM133893), and by the U.S. Department of Energy Office of Biological and Environmental Research (KP1607011). As part of the National Synchrotron Light Source II, a national user facility at Brookhaven National Laboratory, work performed at the Center for BioMolecular Structures is supported in part by the U.S. Department of Energy, Office of Science, Office of Basic Energy Sciences Program under contract number DE-SC0012704. The authors thank ALSAC for support; St. Jude X-ray Center for technical assistance; and Cameron D. Buchman, Wenwei Lin, Jingheng Wang, and other members of the Chen laboratory for valuable discussions.

## Author contributions

E.G.-M., A.D.H., S.C.C., S.N., Y.L. and T.C. conceived and organized the project. E.G.-M., A.D.H., S.N., Y.L., J.W. and S.P. designed and performed the experiments and analyzed data. S.C.C., D.J.M. and J.S. analyzed data. A.D.H., S.N., S.C.C., E.G.-M., Y.L. and T.C. wrote the manuscript with input from all authors. All authors reviewed the final manuscript.

## Competing interests

Authors Taosheng Chen and Yongtao Li declare the following competing financial interest(s): The authors have the following patents related to compounds in this manuscript: (1) Chen T, Lin W, Wang Y. 1,4,5-Substituted 1,2,3-Triazole Analogs as Antagonists of the Pregnane X Receptor. International Patent Application published as WO/2017/165139, 2017; US Patent Application published as US 2019/0077770 A1, 2019. US patent No. 10,550,091 B2 issued, 2020. (2) Chen T, Li Y, Lin W. Small Molecule Modulators of Human Pregnane X Receptor. US Provisional Application No. 63/333,929. Filing date: April 22, 2022. The remaining authors declare no competing interests.
