## [Peer Review File · Nature Communications]

REVIEWER COMMENTS

Reviewer #1 (Remarks to the Author):

Pregnane X receptor (PXR), functioning as a xenobiotic sensor, can be activated by a wide array of structurally diverse compounds and plays a vital role in drug metabolism and drug-drug interactions by regulating transcription of genes related to drug metabolizing enzymes, drug-conjugating enzymes, and drug transporters. Activation or inhibition of PXR by small molecules can affect drug efficacy and/or safety. This group previously identified SPA70 as the first potent and selective PXR antagonist. Subsequent investigations into structure-activity relationships uncovered that minor chemical modifications could transform SPA70 into an agonist, and further derivatization efforts resulted in the development of potent compounds falling into both agonist and antagonist categories, the mechanism of which is unknown. In the current study, the authors reported the first structures of the PXR LBD in conjunction with four antagonists and two structurally related agonists, providing insights into a comprehensive mechanism distinguishing agonism from antagonism. The authors did a great job in elucidating the moieties required for the actions of agonists or antagonists based on the crystal structure data and provided comprehensive comparisons between different nuclear receptors and different species of PXR using a series of agonists and antagonists. The study is novel, and the major conclusions were supported by experimental data.

Specific comments:

1. The functional characterizations of the compounds largely depended on cell culture work, including those using primary hepatocytes. Have any of these new antagonists been tested in vivo? If not, this would be a major limitation that needs to be discussed.
2. The authors mentioned that SJPYT-310 is a more potent PXR antagonist than SPA70. However, in human primary hepatocyte experiment (supplementary Figure 3), when SJPYT-310 was used in combination of rifampicin, it did not show obvious stronger potency than SPA70 in attenuating the rifampicin effect. Could the authors comment on that?
3. Would SJPYT-310 be a potentially better drug candidate than SPA70 in clinical use? Please discuss in more details or comment on the properties of SJPYT-310 besides potency.
4. It would be more convincing if the authors also tested those antagonists in mouse primary hepatocyte together with PCN (referring to supplementary Figure 3).

Reviewer #2 (Remarks to the Author):

Garcia-Maldonado et al. report crystal structures of pregnane X receptor (PXR) in complex with its agonists and antagonists to reveal molecular mechanisms of antagonism and selectivity. They find that antagonists induce PXR conformational changes that would not support transcriptional coactivator recruitment. These results may guide development of effective antagonists to prevent PXR-mediated metabolic problems.

The crystallographic studies are technically sound for defining small-molecule binding. The crystals diffracted reasonably well and structure determination by molecular replacement was straightforward as the compound density showed up as new with expected size and shape, with at least one map refined to near 2Å.

Some minor comments:

1. There is a typo in Supplementary Table 1. Resolution range (Å) 53.44 - 2.02 (2.28 -2.20)a. The highest resolution shell is 2.28 -2.20, the refinement range cannot be 53.44 - 2.02.
2. In Supplementary Figure 8, most compound density does not look as good as nearby protein density (for example, in panel G, the compound density does not look like 2.14Å; not sure that is true or not when looking at the maps or just the figure issue), but compound B factors are similar to or even slightly lower than the average B factors for the protein (water B factors are also relatively low).

Reviewer #3 (Remarks to the Author):

In this study, the authors investigated structural differences in the binding modes of a set of ligands targeting the Pregnane X Receptor. The transformation of a ligand from agonist to antagonist is clearly elucidated through multiple crystal structures under varying conditions. These findings hold significance not only in the PXR field but also serve as an illustrative example of how subtle changes can induce a completely different behavior in the protein mechanism. The methodology appears robust, and the results are sound. While the study admirably explores structural differences in ligands targeting the Pregnane X Receptor and effectively elucidates the transformation from agonist to antagonist through diverse crystal structures, there is an opportunity to enhance the computational aspect. The current computational section lacks sufficient detail and analysis, and it would benefit from a more comprehensive emphasis to provide a deeper understanding of the results.

- While MD simulations align with the proposed mechanism of agonism/antagonism, it is noteworthy that only two ligands from the extensive set have been studied. Given the relatively short duration of simulations (200 ns, achievable in one or two days), it would be valuable to extend MD simulations to include the second agonist and at least one additional antagonist to ensure result consistency.
- The analysis of MD simulations is currently limited to visual inspection and monitoring the distance between two atoms. However, additional insights can be gleaned from these simulations. I recommend exploring per-residue energy contributions for the AF2 binding, using methods such as MM-GBSA. This would provide information on the residues, including the ligand, that predominantly stabilize AF2 binding in different systems, shedding light on the roles of the ligand and other residues.
- The Methods section lacks clarity regarding whether a proper equilibration protocol was implemented before simulation (e.g., a few ns with restrained protein backbone). This is typically a prerequisite, as skipping this step can lead to artifacts. The authors should expand the sentence, "The option to relax systems before simulations was selected," especially for researchers unfamiliar with Desmond options.

Other minor suggestions:

- In the abstract (lines 22-24), the sentence is unclear about who is "represented." I propose restructuring the sentence, such as: "Unfortunately, ligand promiscuity causes approximately two-thirds of receptors to remain clinically untargeted. An example of such a receptor is the pregnane X receptor (PXR), which has evolved to be activated by..."
- On page 1, line 43, the term 'maladies' is somewhat antiquated. Please replace it with a simpler and more commonly used term, such as "disease".

Considering the valuable insights gained from the structural analysis and acknowledging the potential improvements in the computational section, I recommend accepting the manuscript for publication in Nature Communications following the incorporation of the suggested revisions.

Reviewer #4 (Remarks to the Author):

In this manuscript, Garcia-Maldonado et al. report a structure-activity relationship (SAR) and structure-function analysis of the nuclear receptor PXR bound to agonists and antagonists. While the mechanism of agonism is partially understood from crystal structures of the PXR ligand-binding domain (LBD) bound to agonists, it remains poorly understood how to design PXR antagonists.

The authors solved a crystal structure of antagonist-bound PXR LBD, the first reported thus far in the field, and compared the ligand binding mode to an existing agonist-bound structure of similar scaffold. Of note was a 4-methoxy group in the agonist that extends towards the AF-2 helix 12 that makes favorable contacts with several hydrophobic residues; the antagonist ligand does not have this chemical

group and thus does not make favorable contacts. This structure provided a potential molecular blueprint to design antagonists or agonists that either lack or contain a hydrophobic moiety extended towards AF-2 helix 12. Given this hypothesis, the authors synthesized additional compound analogs that contain or lack a 4-methoxy group. Cellular transcription profiling studies show that 4-methoxy compounds behave as PXR agonists, whereas the other analogs lacking this group are PXR antagonists. Specificity assays demonstrate the molecules are specific for PXR and do not bind or activate several other nuclear receptors including mouse PXR, which interestingly shows several amino acid changes in the ligand-binding pocket compared to the human isoform that explain the lack of activity.

Inspection of the crystal structures revealed the antagonists disrupted favorable contacts between the AF-2 helix 12 and the bound SRC-1 coactivator peptide. Molecular dynamics (MD) simulations support this concept and show that the AF-2 helix 12 in antagonist-bound PXR LBD pushes outward in a conformation that would overlap with the bound SRC-1 peptide, providing a plausible mechanism by which the antagonists block PXR activity.

The authors further validated the mechanism of antagonism using structure-guided mutagenesis where certain amino acid mutations converted the PXR antagonists into agonists. A crystal structure of one mutant protein (L428V) provided a structural rationale for the activity change: the mutation reorients the antagonist in the pocket relative to wild-type PXR resulting in more favorable contacts with the AF-2 helix 12.

Overall, this is a very thorough study with experiments and results described in a rigorous manner. If I were to highlight one weakness, there is considerable detail provided describing the different contacts between the various ligands and PXR LBD residues, which may detract from a broader readership; however, this is also a strength because the authors do an outstanding job of describing structure-guided hypothesis for designing PXR antagonists and agonists as well as the subsequent testing of the hypothesis tested using medicinal chemistry, crystallography, and biochemical/cellular assays.

The authors refer to the "AF-2 helix" throughout the manuscript. In the nuclear receptor field, the "AF-2" is understood to be a 3D surface composed of three structural elements in the LBD: helix 3, helix 5, and helix 12. The authors' use of "AF-2 helix" likely refers to helix 12, and I would therefore recommend the authors use different phrasing such as "AF-2 helix 12" to call out that specific structural element.

Could the analysis of the existing MD data be extended using methods such as RMSF or probability distance histogram distributions to further characterize the impact of agonist vs antagonist on AF-2 helix 12 positioning/dynamics? This would compliment the MD trajectory snapshots shown implicating a mechanism where antagonists push AF-2 helix 12 outward to clash with the coactivator peptide.

We thank the reviewers for their thorough evaluation of all technical areas of our manuscript and the professional quality with which the comments were presented. We have edited the text and performed additional experiments and analysis to address the comments, and our specific responses are written below in green text.

Reviewer #1:

Pregnane X receptor (PXR), functioning as a xenobiotic sensor, can be activated by a wide array of structurally diverse compounds and plays a vital role in drug metabolism and drug-drug interactions by regulating transcription of genes related to drug metabolizing enzymes, drug-conjugating enzymes, and drug transporters. Activation or inhibition of PXR by small molecules can affect drug efficacy and/or safety. This group previously identified SPA70 as the first potent and selective PXR antagonist. Subsequent investigations into structure-activity relationships uncovered that minor chemical modifications could transform SPA70 into an agonist, and further derivatization efforts resulted in the development of potent compounds falling into both agonist and antagonist categories, the mechanism of which is unknown. In the current study, the authors reported the first structures of the PXR LBD in conjunction with four antagonists and two structurally related agonists, providing insights into a comprehensive mechanism distinguishing agonism from antagonism. The authors did a great job in elucidating the moieties required for the actions of agonists or antagonists based on the crystal structure data and provided comprehensive comparisons between different nuclear receptors and different species of PXR using a series of agonists and antagonists. The study is novel, and the major conclusions were supported by experimental data.

Specific comments:

- The functional characterizations of the compounds largely depended on cell culture work, including those using primary hepatocytes. Have any of these new antagonists been tested in vivo? If not, this would be a major limitation that needs to be discussed.
 - We have now noted this limitation in the discussion section: *“Notably, these ligands display high potency with no noticeable toxicity and are selective for PXR over a panel of related NRs, although these properties are yet to be evaluated in vivo.”*
- The authors mentioned that SJPYT-310 is a more potent PXR antagonist than SPA70. However, in human primary hepatocyte experiment (supplementary Figure 3), when SJPYT-310 was used in combination of rifampicin, it did not show obvious stronger potency than SPA70 in attenuating the rifampicin effect. Could the authors comment on that?
 - PHH from different donors are highly variable in PXR activity and induction capacity; this is evident in the rifampicin induction of our three donors. Accordingly, in our view, PHH provide qualitative, rather than quantitative, data. More donors along with more replicates must be assessed to make conclusions about potency in this system, and we have refrained from making quantitative statements in the text.
- Would SJPYT-310 be a potentially better drug candidate than SPA70 in clinical use? Please discuss in more details or comment on the properties of SJPYT-310 besides potency.
 - Because our study is focused on the structural basis of PXR antagonism, we do not have data to describe clinical properties. Studies of such nature (PK/PD, metabolic stability, etc.) will be conducted in the future. Furthermore, a drug candidate should be well characterized. SJPYT-310, but not SPA70, has now been co-crystallized with PXR. Therefore, from a compound characterization perspective, SJPYT-310 is a potentially better drug candidate than SPA70 for future clinical development.

- It would be more convincing if the authors also tested those antagonists in mouse primary hepatocyte together with PCN (referring to supplementary Figure 3).
 - We have added mouse hepatocytes to Supplementary Figure 3b. The data are consistent with Supplementary Figures 4 & 5 that show the compounds do not activate or inhibit mouse PXR. This statement has been added to the main text: *“The compounds were highly selective for human PXR, with no observed activation or inhibition of other receptors and no mPXR modulation in primary mouse hepatocytes (Figure 3a, Supplementary Figures 3b, 4, and 5).”*

Reviewer #2:

Garcia-Maldonado et al. report crystal structures of pregnane X receptor (PXR) in complex with its agonists and antagonists to reveal molecular mechanisms of antagonism and selectivity. They find that antagonists induce PXR conformational changes that would not support transcriptional coactivator recruitment. These results may guide development of effective antagonists to prevent PXR-mediated metabolic problems.

The crystallographic studies are technically sound for defining small-molecule binding. The crystals diffracted reasonably well and structure determination by molecular replacement was straightforward as the compound density showed up as new with expected size and shape, with at least one map refined to near 2Å.

Some minor comments:

1. There is a typo in Supplementary Table 1. Resolution range (Å) 53.44 - 2.02 (2.28 -2.20)a. The highest resolution shell is 2.28 -2.20, the refinement range cannot be 53.44 - 2.02.
 - We have included the correct resolution range (Å) 53.44 - 2.20 (2.28 -2.20) in Supplementary Table 1. Thank you for noticing this error.
2. In Supplementary Figure 8, most compound density does not look as good as nearby protein density (for example, in panel G, the compound density does not look like 2.14Å; not sure that is true or not when looking at the maps or just the figure issue), but compound B factors are similar to or even slightly lower than the average B factors for the protein (water B factors are also relatively low).
 - We have replaced some panels showing clear densities for compounds including panel G in Supplementary Figure 8. The B factors for most of the compounds are similar to or slightly lower than the average B factors for the protein, suggesting that compounds fit well into the densities.

Reviewer #3:

In this study, the authors investigated structural differences in the binding modes of a set of ligands targeting the Pregnane X Receptor. The transformation of a ligand from agonist to antagonist is clearly elucidated through multiple crystal structures under varying conditions. These findings hold significance not only in the PXR field but also serve as an illustrative example of how subtle changes can induce a completely different behavior in the protein mechanism. The methodology appears robust, and the results are sound. While the study admirably explores structural differences in ligands targeting the Pregnane X Receptor and effectively elucidates the transformation from agonist to antagonist through diverse crystal structures, there is an

opportunity to enhance the computational aspect. The current computational section lacks sufficient detail and analysis, and it would benefit from a more comprehensive emphasis to provide a deeper understanding of the results.

1. While MD simulations align with the proposed mechanism of agonism/antagonism, it is noteworthy that only two ligands from the extensive set have been studied. Given the relatively short duration of simulations (200 ns, achievable in one or two days), it would be valuable to extend MD simulations to include the second agonist and at least one additional antagonist to ensure result consistency.
 - We have now included an additional agonist (SJPYT-326) and antagonist (SJPYT-312) in the MD analysis, with consistent results – Supplementary Figure 11.
2. The analysis of MD simulations is currently limited to visual inspection and monitoring the distance between two atoms. However, additional insights can be gleaned from these simulations. I recommend exploring per-residue energy contributions for the AF2 binding, using methods such as MM-GBSA. This would provide information on the residues, including the ligand, that predominantly stabilize AF2 binding in different systems, shedding light on the roles of the ligand and other residues.
 - We have added analysis of protein-ligand contacts over the course of the four simulations and found that the AF2 residue L428 is a stabilizing interaction for agonists – Supplementary Figure 12. This is also now discussed in the results section: “Residue interaction analysis shows agonist-biased (L239, L411, and L428) and antagonist-biased (I254, H407, and F420) interactions, with L428 being the key α 12 interaction for agonists (Supplementary Figure 12).”
3. The Methods section lacks clarity regarding whether a proper equilibration protocol was implemented before simulation (e.g., a few ns with restrained protein backbone). This is typically a prerequisite, as skipping this step can lead to artifacts. The authors should expand the sentence, “The option to relax systems before simulations was selected,” especially for researchers unfamiliar with Desmond options.
 - We have added additional methods details. We have also added C α RMSD plots to show that the proteins are well-equilibrated during the production runs – Supplementary Figure 11f.

Other minor suggestions:

1. In the abstract (lines 22-24), the sentence is unclear about who is “represented.” I propose restructuring the sentence, such as: “Unfortunately, ligand promiscuity causes approximately two-thirds of receptors to remain clinically untargeted. An example of such a receptor is the pregnane X receptor (PXR), which has evolved to be activated by...”
 - We have edited the sentence to read as suggested.
2. On page 1, line 43, the term ‘maladies’ is somewhat antiquated. Please replace it with a simpler and more commonly used term, such as “disease”.
 - “Maladies” has been replaced with “diseases.”

Considering the valuable insights gained from the structural analysis and acknowledging the potential improvements in the computational section, I recommend accepting the manuscript for publication in Nature Communications following the incorporation of the suggested revisions.

- Thank you for the support of our manuscript and the constructive feedback for the MD.

Reviewer #4:

In this manuscript, Garcia-Maldonado et al. report a structure-activity relationship (SAR) and structure-function analysis of the nuclear receptor PXR bound to agonists and antagonists. While the mechanism of agonism is partially understood from crystal structures of the PXR ligand-binding domain (LBD) bound to agonists, it remains poorly understood how to design PXR antagonists.

The authors solved a crystal structure of antagonist-bound PXR LBD, the first reported thus far in the field, and compared the ligand binding mode to an existing agonist-bound structure of similar scaffold. Of note was a 4-methoxy group in the agonist that extends towards the AF-2 helix 12 that makes favorable contacts with several hydrophobic residues; the antagonist ligand does not have this chemical group and thus does not make favorable contacts. This structure provided a potential molecular blueprint to design antagonists or agonists that either lack or contain a hydrophobic moiety extended towards AF-2 helix 12. Given this hypothesis, the authors synthesized additional compound analogs that contain or lack a 4-methoxy group. Cellular transcription profiling studies show that 4-methoxy compounds behave as PXR agonists, whereas the other analogs lacking this group are PXR antagonists. Specificity assays demonstrate the molecules are specific for PXR and do not bind or activate several other nuclear receptors including mouse PXR, which interestingly shows several amino acid changes in the ligand-binding pocket compared to the human isoform that explain the lack of activity.

Inspection of the crystal structures revealed the antagonists disrupted favorable contacts between the AF-2 helix 12 and the bound SRC-1 coactivator peptide. Molecular dynamics (MD) simulations support this concept and show that the AF-2 helix 12 in antagonist-bound PXR LBD pushes outward in a conformation that would overlap with the bound SRC-1 peptide, providing a plausible mechanism by which the antagonists block PXR activity.

The authors further validated the mechanism of antagonism using structure-guided mutagenesis where certain amino acid mutations converted the PXR antagonists into agonists. A crystal structure of one mutant protein (L428V) provided a structural rationale for the activity change: the mutation reorients the antagonist in the pocket relative to wild-type PXR resulting in more favorable contacts with the AF-2 helix 12.

Overall, this is a very thorough study with experiments and results described in a rigorous manner. If I were to highlight one weakness, there is considerable detail provided describing the different contacts between the various ligands and PXR LBD residues, which may detract from a broader readership; however, this is also a strength because the authors do an outstanding job of describing structure-guided hypothesis for designing PXR antagonists and agonists as well as the subsequent testing of the hypothesis tested using medicinal chemistry, crystallography, and biochemical/cellular assays.

- We thank the reviewer for this thoughtful comment. We have made minor edits to the text to make our manuscript more easily understandable to a broader audience while maintaining the detail required to guide our experimentation.

The authors refer to the “AF-2 helix” throughout the manuscript. In the nuclear receptor field, the “AF-2” is understood to be a 3D surface composed of three structural elements in the LBD: helix 3, helix 5, and helix 12. The authors’ use of “AF-2 helix” likely refers to helix 12, and I would therefore recommend the authors use different phrasing such as “AF-2 helix 12” to call out that specific structural element.

- We have edited occurrences of AF-2 to specifically refer to helix 12 (now labeled $\alpha 12$). The “AF-2” label is still used to refer to the 3D surface.

Could the analysis of the existing MD data be extended using methods such as RMSF or probability distance histogram distributions to further characterize the impact of agonist vs antagonist on AF-2 helix 12 positioning/dynamics? This would compliment the MD trajectory snapshots shown implicating a mechanism where antagonists push AF-2 helix 12 outward to clash with the coactivator peptide.

- We have added analysis of protein-ligand contacts over the course of our four simulations (two original and two more from manuscript revision) and found agonist-biased (L239, L411, and L428) and antagonist-biased (I254, H407, and F420) interactions – Supplementary Figure 12. This is also now discussed in the results section: “*Residue interaction analysis shows agonist-biased (L239, L411, and L428) and antagonist-biased (I254, H407, and F420) interactions, with L428 being the key $\alpha 12$ interaction for agonists (Supplementary Figure 12).*”

REVIEWERS' COMMENTS

Reviewer #1 (Remarks to the Author):

The authors have adequately addressed my comments and as a result, the manuscript has improved from its previous version.

Reviewer #2 (Remarks to the Author):

The authors have carefully addressed my previous comments. I do not have anything additional to add.

Reviewer #3 (Remarks to the Author):

Authors satisfactorily addressed all of my previous comments. I have a few additional suggestions:

In the Methods section, the authors briefly mentioned, 'missing loops were filled using the AlphaFold model of PXR as a guide.' However, more details would be appreciated. Could the authors specify whether this process was conducted using Coot, performed manually, or utilized an external tool?

Regarding the Calpha RMSD plot (Supplementary Figure 11f), the RMSD values appear to be high. This is likely attributable to the significant deviation of the long loop (residues 163-196). To enhance clarity and support the stability of the simulations, I recommend that the authors consider excluding these atoms from the RMSD calculations.

Reviewer #4 (Remarks to the Author):

The authors satisfactorily addressed my comments. The new Supplementary Figure 12 nicely shows the different ligand interactions between the agonists and antagonists.

We again thank the reviewers for their evaluation of our manuscript. We have made minor text edits to address the final comments of Reviewer 3, and our specific responses are written below in green text.

Reviewer #1:

The authors have adequately addressed my comments and as a result, the manuscript has improved from its previous version.

Reviewer #2:

The authors have carefully addressed my previous comments. I do not have anything additional to add.

Reviewer #3:

Authors satisfactorily addressed all of my previous comments. I have a few additional suggestions:

In the Methods section, the authors briefly mentioned, 'missing loops were filled using the AlphaFold model of PXR as a guide.' However, more details would be appreciated. Could the authors specify whether this process was conducted using Coot, performed manually, or utilized an external tool?

- We have edited the methods section to clarify that the loops were filled manually in Coot.

Regarding the Calpha RMSD plot (Supplementary Figure 11f), the RMSD values appear to be high. This is likely attributable to the significant deviation of the long loop (residues 163-196). To enhance clarity and support the stability of the simulations, I recommend that the authors consider excluding these atoms from the RMSD calculations.

- Thank you for evaluating the added data. Please note that the units in this plot are angstroms, not nanometers, and we believe that the observed RMSDs of 2-4 Å (0.2-0.4 nm) are quite reasonable.

Reviewer #4:

The authors satisfactorily addressed my comments. The new Supplementary Figure 12 nicely shows the different ligand interactions between the agonists and antagonists.